# Deep Learning and Machine Learning Models for Landslide Susceptibility Mapping with Remote Sensing Data

Muhammad Afaq Hussain [1], Zhanlong Chen [1,*], Ying Zheng [2], Yulong Zhou [3] and Hamza Daud [4]

1   School of Computer Science, China University of Geosciences, Wuhan 430074, China
2   Ningbo Alatu Digital Science and Technology Corporation Limited, Ningbo 315000, China
3   School of Geography and Information Engineering, China University of Geosciences, Wuhan 430074, China
4   Badong National Observation and Research Station of Geohazards, China University of Geosciences, Wuhan 430074, China
*   Correspondence: chenzl@cug.edu.cn

**Abstract:** Karakoram Highway (KKH) is an international route connecting South Asia with Central Asia and China that holds socio-economic and strategic significance. However, KKH has extreme geological conditions that make it prone and vulnerable to natural disasters, primarily landslides, posing a threat to its routine activities. In this context, the study provides an updated inventory of landslides in the area with precisely measured slope deformation (Vslope), utilizing the SBAS-InSAR (small baseline subset interferometric synthetic aperture radar) and PS-InSAR (persistent scatterer interferometric synthetic aperture radar) technology. By processing Sentinel-1 data from June 2021 to June 2023, utilizing the InSAR technique, a total of 571 landslides were identified and classified based on government reports and field investigations. A total of 24 new prospective landslides were identified, and some existing landslides were redefined. This updated landslide inventory was then utilized to create a landslide susceptibility model, which investigated the link between landslide occurrences and the causal variables. Deep learning (DL) and machine learning (ML) models, including convolutional neural networks (CNN 2D), recurrent neural networks (RNNs), random forest (RF), and extreme gradient boosting (XGBoost), are employed. The inventory was split into 70% for training and 30% for testing the models, and fifteen landslide causative factors were used for the susceptibility mapping. To compare the accuracy of the models, the area under the curve (AUC) of the receiver operating characteristic (ROC) was used. The CNN 2D technique demonstrated superior performance in creating the landslide susceptibility map (LSM) for KKH. The enhanced LSM provides a prospective modeling approach for hazard prevention and serves as a conceptual reference for routine management of the KKH for risk assessment and mitigation.

**Keywords:** convolutional neural network; recurrent neural networks; landslide susceptibility mapping; extreme gradient boosting; random forest

## 1. Introduction

Landslides, one of the most common natural disasters, prevalent in mountainous regions worldwide, pose significant threats to the ecosystem [1]. Landslides are accounted as the downhill movement of debris, soil, and rocks under the force of gravity and can be classified based on the materials involved (mud, rock, soil, or debris) and their movement type (topple flow or slide) [2]. The factors leading to landslides are a combination of tectonics, geomorphology, and climate change, which culminate in a critical slope evolution [3,4]. Other triggering factors contribute to landslides depending on the specific features of the area. Natural variables such as rainfall, rapid snowmelt, earthquakes, and anthropogenic activities, e.g., habitation construction, irrigation, etc., can play a role in the occurrence of landslides [5]. While landslides are often regarded as a natural process, their occurrence mostly has been influenced by anthropogenic activity [6]. In recent years, exponentially

growing populations, a surge in infrastructure development, and settlement growth in developing countries' mountainous regions have increased the probability of landslides, leading to an alarming increase in landslide-related fatalities [7].

The "China-Pakistan Economic Corridor" (CPEC), a significant project under the "One Belt and One Road" initiative, is centered around connecting Pakistan and China via the Karakoram Highway (KKH). The KKH was constructed from 1974 to 1978 and commenced operation in 1979. The highway encompasses most of the route of the CPEC. However, this vital route faces challenges due to the high mountainous terrain with overflowing loose debris and heavy rainfall, triggering frequent and severe geological catastrophes such as glacier debris flows, rock falls, landslides, debris and soil slippage, and avalanches [8]. Determining landslide probabilities along the KKH is a complex process influenced by limited data availability, technical limitations, and harsh environments. Since its completion, the reputation of the KKH has been marred by various geohazards [9]. Specifically, earth-induced landslides in 2005 caused considerable damage to the highway [10]. Enormous rockslides and rock avalanches have occurred, with over 115 incidents reported since 1987 [11]. Moreover, in 2010, a landslide blocked the Hunza River, inundating 19 km of the highway with a loss of 20 lives and damaging 350 houses [12]. The geological conditions along the KKH pose additional challenges, including fragile and weathered rock masses, varying climates, low and high terrains, diverse stratigraphy, and local variations in tectonic motion. Due to these factors, the study region has become a geohazard laboratory. Enhancing precise LSM along the KKH to mitigate the risks posed by these natural hazards is imperative.

Recently, remote sensing (RS) and geographic information systems (GIS) technology have made remarkable technological progress. The utilization of GIS spatial analysis tools and remote-sensing-derived data has enhanced the effectiveness of landslide susceptibility mapping for accurate assessment. Here, comprehensive landslide inventory data and knowledge of landslide conditioning factors are crucial for both data-driven spatial modeling and knowledge-based approaches [13]. Researchers have conducted numerous studies using bivariate analyses to quantify the spatial correlations between landslides and specific factors that influence their dispersion [14–17]. Several other studies have applied knowledge-based spatial approaches to produce natural risk vulnerability maps, fuzzy logic models [18,19], the analytical hierarchy process (AHP) [20], and the evidential belief function [21], as well as data-driven spatial approaches such as support vector machines [22–24], logistic regression methods [24,25], artificial neural network (ANN) models [26–28], alternating decision tree (ADTree) [29], principal component analysis (PCA) [30], deep belief network (DBN) [31], decision tree [25,32], superposable neural networks [33], and naïve Bayes [34]. Expertise-based models often encounter challenges due to their reliance on expert opinions, which can introduce biases [35,36].

The primary strengths of probabilistic and ML approaches lie in their objective statistical foundation, consistency, capacity for precisely analyzing the factors influencing landslide development, and capacity building for updates. In this perspective, researchers are continuously seeking new and relatively more robust algorithms that can generalize across different spatial scales [37,38]. Deep learning algorithms, which are specifically developed for large datasets but have seen limited application thus far, need to be implemented and evaluated in this context. Currently, deep learning models, particularly recurrent neural networks and convolutional neural networks, have demonstrated remarkable success across various applications, making them well suited for handling big data [39]. RNNs, like other DL models, comprise a loss function, learnable parameters, and layers [40]. On the other hand, CNNs differ from RNNs as they include convolutional and pooling layers and focus solely on the current input data, while RNNs consider both the earlier provided inputs and present input data [41]. CNNs have proven effective in tasks like semantic segmentation and object detection [7]. Conversely, RNNs show superior performance in tasks such as image recognition, characterization, and sequential data analysis, including time series spatial data [42]. Despite the acceptable results achieved by CNNs and RNNs in

various domains, their true efficiency and capabilities in landslide modeling and large-scale landslide susceptibility mapping (LSM) on big data have not been thoroughly analyzed [13]. A few deep learning models have been utilized for natural hazard vulnerability mapping, containing landslide susceptibility mapping and flash floods [43–45]. However, these studies have separately employed different deep learning models, and their relative proficiency has not been evaluated yet.

In recent years, interferometric synthetic aperture radar (InSAR) methods have acquired universal approval and usage as tools for landslide monitoring and mapping. Over the past two decades, the RS technique, particularly In-SAR, has demonstrated substantial possibility across different fields, including the study of landslide deformation [46] and groundwater extraction [47]. PS-InSAR proves useful in automatic slow-moving landslide mapping using a spatial statistical technique, the detection of particular landslides and the delineation of extended unstable regions, redefining of the limits of historical landslides, the detection of landslides using a multitemporal analysis of SAR imagery, and the verification of the terrain elements causing slope deformation [48]. In areas prone to frequent and rapid large landslides, RS provides a solution through surveys and advanced detection methods [49]. These techniques can greatly aid in assessing and creating landslide inventory maps. Various methods of InSAR have been effectively used in mapping slope displacement, including that in [50], the assessment of land displacement places identified by using SBAS-InSAR [51], the D-InSAR technique for landslide observing and land deformation [51,52], the coherence pixel technique [53], the SqeeInSAR approach to measuring surface motion [51], interferometric point target analysis [54], the use of StaMPS to evaluate the displacement in a high-vegetation region [55,56], and the PSInSAR method to compute the movement of landslides. These approaches are related to detecting and mapping landslide events, as mentioned in [54,57,58].

In this study, a combination of optical RS analysis and the InSAR technique is utilized to identify landslides and create an updated landslide inventory. The main goals are as follows: (1) mapping all types of landslides along the KKH and estimating displacement maps to identify new landslides, identify unstable places, and redefine the boundaries of previously identified landslides based on the deformation model; (2) generating a landslide susceptibility map using state-of-the-art ML and deep learning (DL) models, including random forest, XGBoost, recurrent neural networks, and convolutional neural networks; (3) comparing the performance of these advanced ML and DL models in terms of landslide susceptibility; (4) assessing the significance and relationships of environmental and anthropogenic factors influencing landslides and their role in evaluating landslide susceptibility in the study area; and (5) determining the most accurate susceptible model reliant on precision and AUC value. Despite the fact that the KKH faces significant landslide threats every summer, previous research has not adequately addressed the issue. Therefore, the landslide susceptibility map produced in this study will aid urban planning and disaster reduction efforts in the area. Moreover, the final InSAR-based landslide inventory will assist in tracking risky areas to minimize future hazards and fatalities. It is imperative to highlight that no previous studies have applied RNNs and CNNs for LSM at KKH. As the first study to utilize and compare these ML and DL models for LSM in this region, it will substantially contribute to the scientific literature.

## 2. Materials and Methods

### 2.1. Study Area and Geological Settings

The KKH in northern Pakistan is significant as part of the CPEC but is prone to frequent disruptions caused by various geological and hydro-climatological hazards. The study area was focused along a 263 km section of the KKH, passing through different districts of Gilgit Baltistan. A 5 km buffer around this section, covering an area of 3320 km$^2$ (Figure 1), was examined for the study. The terrain in the study area is rugged, with elevations ranging from 822 to 5545 m above mean sea level. The area experiences mild summers and harsh winters, and the yearly rainfall varies from 120 to 130 mm. The minimum and maximum

temperatures are −21 °C and 16 °C, respectively, according to data from the Meteorological Department of Pakistan (https://www.pmd.gov.pk, accessed on 15 March 2023).

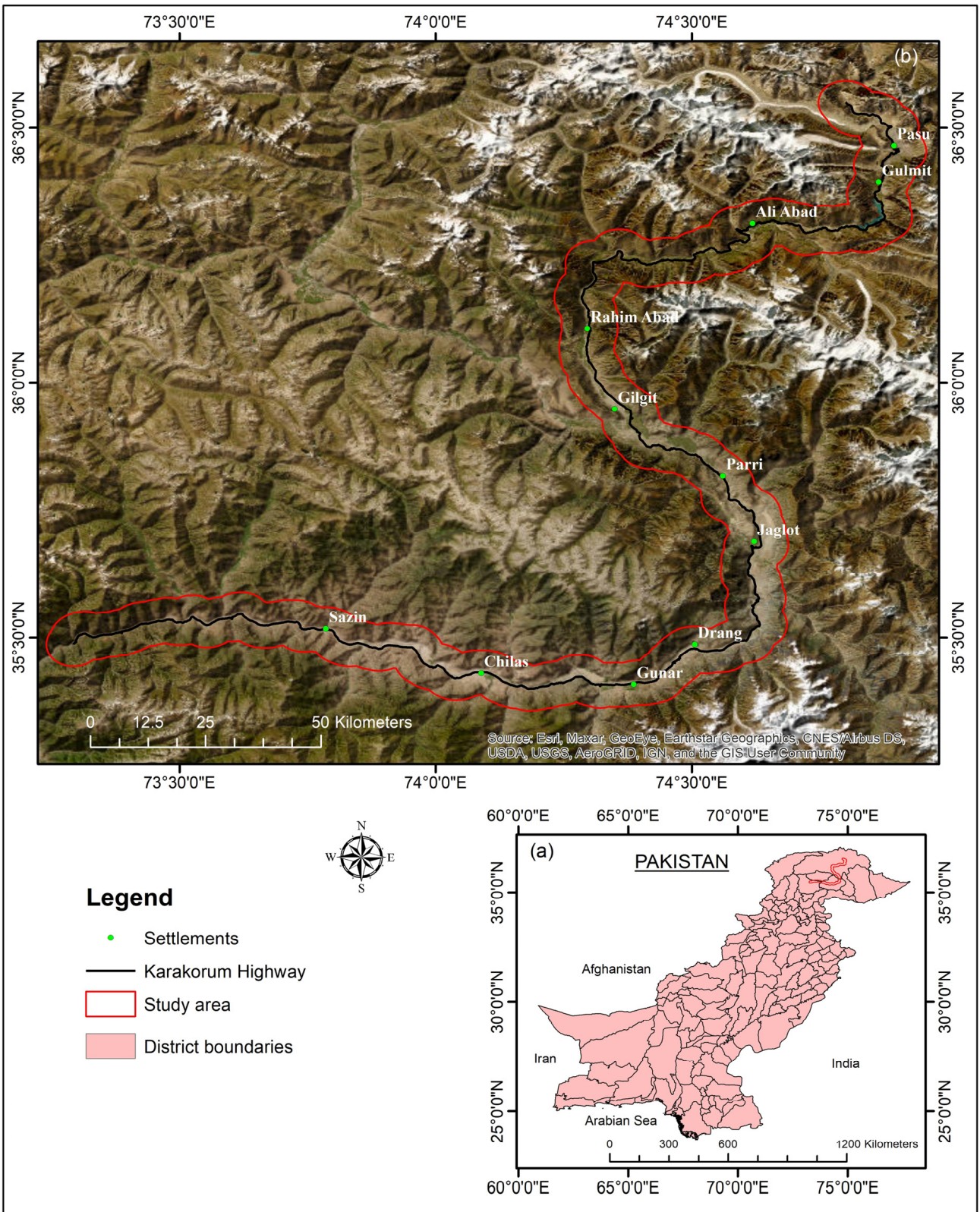

**Figure 1.** Location of the study area. (**a**) Pakistan, (**b**) Study area in red outline.

The lithology in the research area is significant in triggering landslides. The rocks of the area are primarily of Mesozoic and Paleozoic age. Based on the geological map produced by Searle et al. [59], the research area comprises a diverse range of rocks, including sedimentary, volcano-sedimentary, igneous, volcanic, and metamorphic rocks (Figure 2). These rocks are further stratified into various types, such as greenschist, siliciclastic, carbonates, basalt, andesite, granite, gabbro, and others. The Chalt schists, kilk formation, Quaternary sediments, deformed Misgar slates, and Gujhal dolomite are the most significant among the area's lithologic formations. All of these lithologies have been tectonically affected and have contributed to slope destabilization along the highway [9]. Over time, the lithologies exposed along the KKH have undergone weathering and weakening due to anthropogenic, hydro-climatic, and seismic events, resulting in significant landslides and surface distortion. Structurally, the area is sophisticated because it lies in a convergent boundary, specifically the Main Karakoram Thrust. This structural setting adds to the susceptibility of the area to geological hazards like landslides.

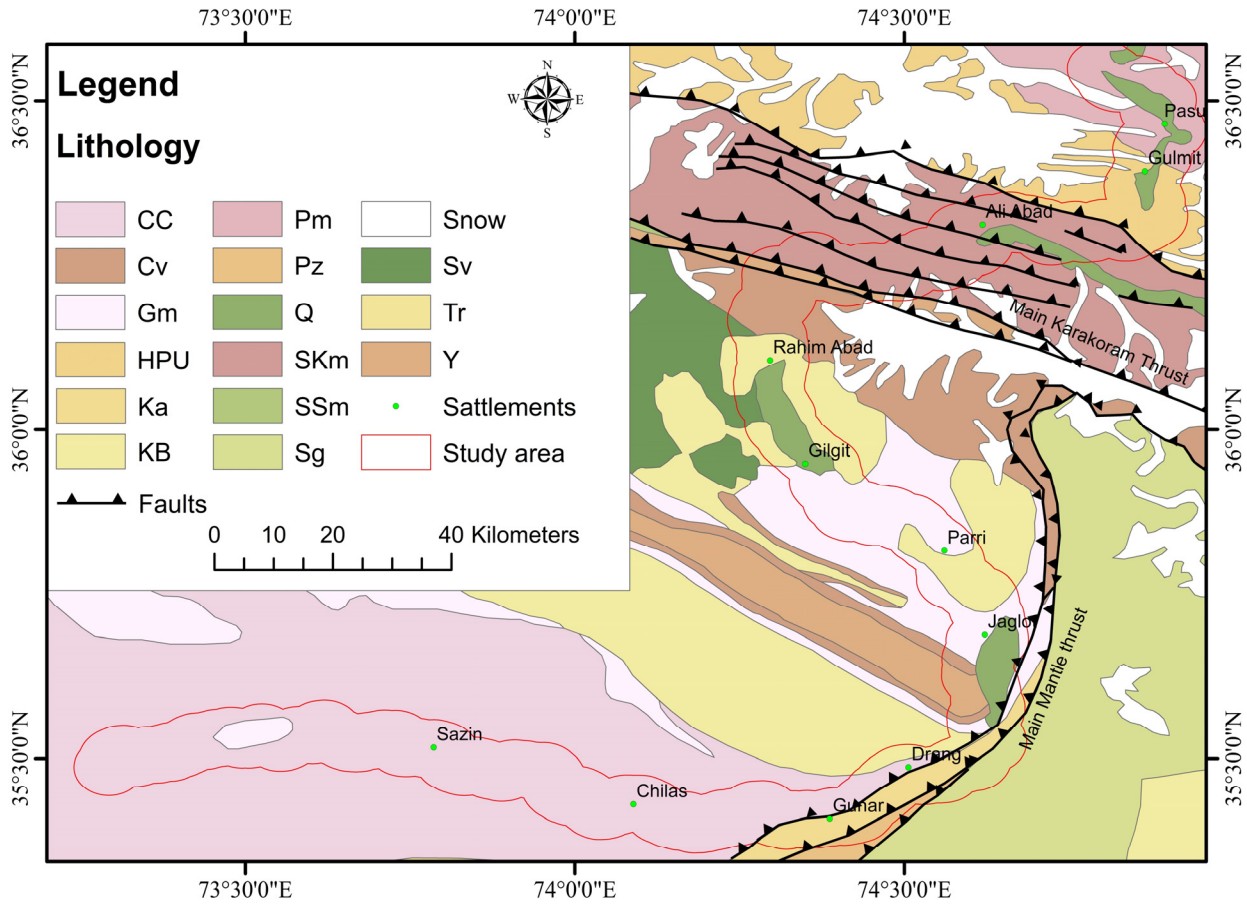

**Figure 2.** Regional geological map of the study region, which depicts the fault lines (MKT and MMT) and Geological units in the research area, where CC is Chilas Complex (Mafic and Ultra-mafic rocks), Gm is Gilgit complex metasedimentary rocks, Cv is Chalt group, Pm stands for Permian massive Limestone, HPU is a Hunza plutonic unit, Ka stands for Komila amphibolite Complex, Sg is Sumayar Leucogranite, KB is Kohistan Batholiths, SSm is Shyok Suture Melange, Q is Quaternary deposits, SKm stands for southern Karakoram complex, Pz is Palaeozoic Metasedimentary rocks, Sv is Kohistan Arc sequence, Tr is Triassic massive dolomite and limestone, and Y stands for Yasin group.

### 2.2. Datasets

The Alaska Satellite Facility (ASF) datasets provide an ALOS-PALSAR DEM (digital elevation model) with a resolution of 12.5 m, which was accessed from https://search.asf.alaska.edu/ (accessed on 10 February 2023). Additionally, Sentinel-2 images with a

resolution of 10 m were derived from the Copernicus dataset (https://scihub.copernicus.eu, accessed on 10 February 2023) to produce a landcover map for the study area. Geological maps and fault lines for the research area were processed using the ArcGIS software 10.8 to understand the geological features [59,60]. To assess the relationship between rainfall and landslide events, annual precipitation data were obtained from the GIOVANNI online database system (https://giovanni.gsfc.nasa.gov/giovanni/, accessed on 15 March 2023), as rainfall and landslides are found to be directly proportionally related. For this study, two years (June 2021 to June 2023) of C-band Sentinel-1A SAR dataset imagery was obtained from the ASF (search.asf.alaska.edu) online system. The dataset contains scenes in descending and ascending tracks, as presented in Table 1. Figure 3 depicts the technical route of the research.

**Table 1.** Datasets used in PS-InSAR and SBAS-InSAR analysis.

| Data Information | Ascending | Descending |
|---|---|---|
| Product type | Sentinel 1 SLC | |
| Acquisition mode | IW | |
| Polarization | VV | |
| Wavelength (m) | 0.056 | |
| No of images | 63 | 60 |
| Time period | June 2021–June 2023 | |
| Frame | 114 | 473 |
| Track | 100 | 107 |
| Coverage (km$^2$) | 250 | |
| Incident angle | Horizontal (~45°) to vertical (~23°) | |
| Azimuth resolution and range (m) | 5 × 20 | |

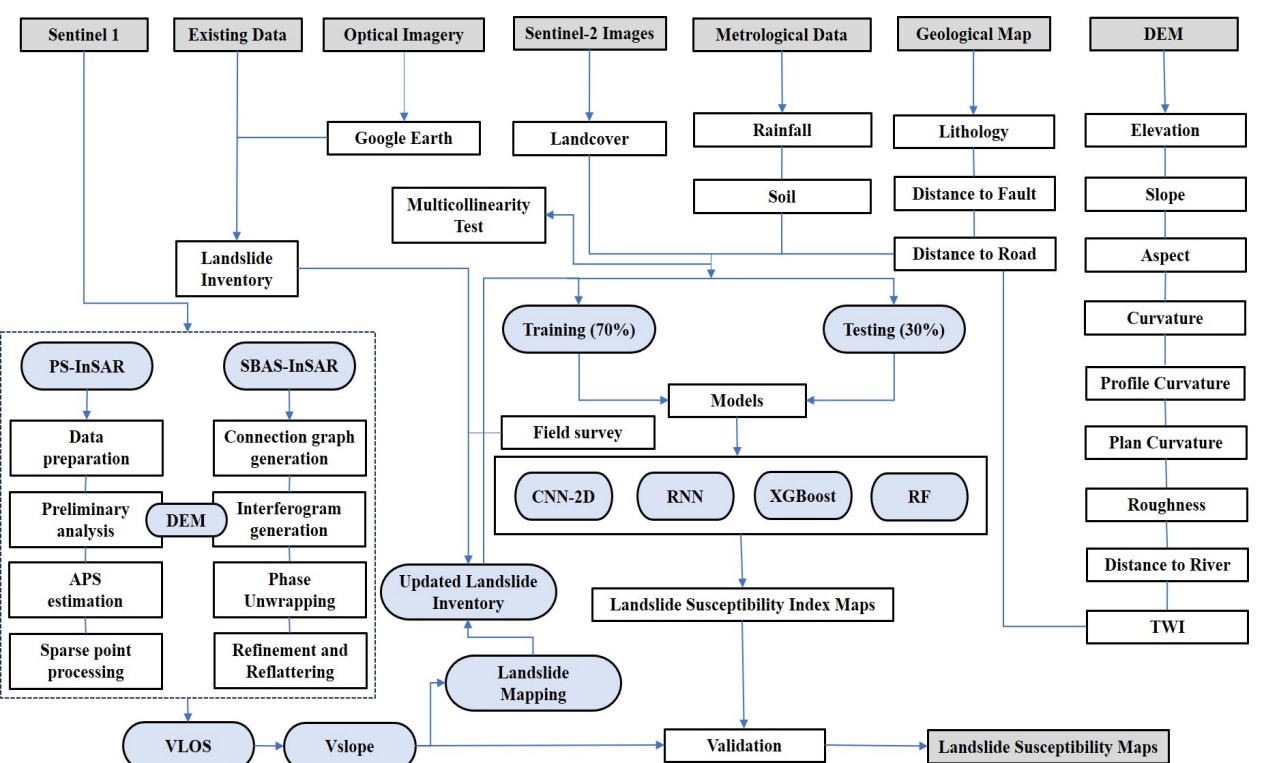

**Figure 3.** Technical route applied in the research.

*2.3. Updated Landslide Inventory*

Creating landslide inventory maps is a crucial step in LSM [61]. These maps provide essential information about the landscape's locations and types of landslides, serving as a foundation for predicting future landslides [1]. Landslide inventory maps are generated for a diversity of reasons, including identifying the type and location of landslides in a particular region; showing the impact of a single landslide-triggering incident, such as a rapid snowmelt incident, an intense rainfall event, or an earthquake; emphasizing the quantity of mass movements; calculating the frequency area statistics of slope failures; and providing relevant data to build landslide risk models or susceptibility models [62].

In this study, a total of 571 landslides were mapped using various techniques, including SBAS-InSAR and PS-InSAR, past studies [12,60,63], Frontier Works Organization road clearance logs, optical imagery analysis, Google Earth, and fieldwork in the study area. The landslide inventory, on the other hand, was developed through the visual interpretation of Sentinel-2 images with 10 m resolution (2022) and by using Google Earth, and it was checked using previous documents and a field evaluation of the study region. The inventory map was categorized into eight categories based on the material displacement, comprising 99 scree, 113 rockslides, 28 rock falls, 20 rock avalanches, 20 debris slides, 271 debris flows, 7 debris falls, and 13 complex slides (Figure 4).

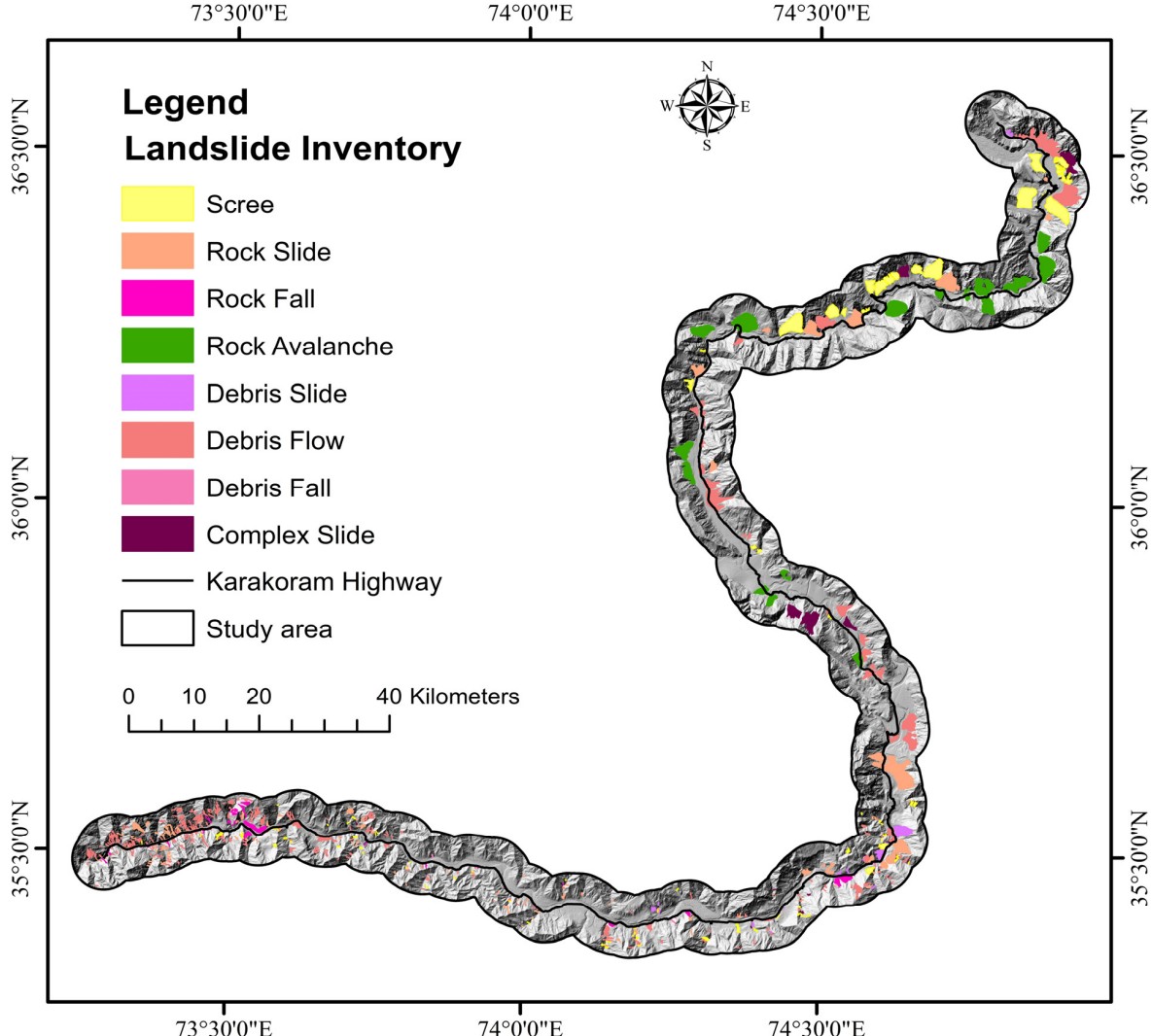

**Figure 4.** Landslide inventory categorization derived from movement along KKH (black line) with various colors.

The inventory map is applied to verify the identification of landslides through InSAR in the study area. Following the InSAR analysis, potential landslides are recognized based on their high displacement velocity, and these newly detected landslides are then incorporated into the updated version of the inventory map.

### 2.4. Landslide Conditioning Factors (LCFs)

The process of achieving high accuracy in the landslide susceptibility model and predicting vulnerable areas heavily relies on carefully selecting and preparing the Landslide Conditioning Factors (LCFs) database [64]. There are no universal standards for selecting independent variables for LSM [8,65,66]. The LCFs were chosen in this study based on information gathered from the relevant literature, data specific to the study area, and field investigations. Fifteen LCFs were selected for the current study, including slope, aspect, topographic wetness index (TWI), distance to roads, lithology, distance to rivers, roughness, distance to faults, curvature, precipitation, plan curvature, soil, profile curvature, elevation, and landcover (Table 2). Thematic layers with a spatial resolution of 12.5 × 12.5 m pixel size were prepared (Figures 5 and 6), all using the WGS84 Datum, UTM-Zone 43 coordinate system.

**Table 2.** List of landslide causative variables used in the research.

| S.NO | Variables | Sources | Description/Extraction |
|---|---|---|---|
| 1 | Aspect, Elevation, Slope, Curvature, Plan Curvature, Profile Curvature, TWI, Distance to River, Roughness, | Digital Elevation Model | ALOS-PALSAR-DEM (https://search.asf.alaska.edu, accessed on 10 February 2023) |
| 2 | Lithology, Distance to Fault, Distance to road | Geological Map | Geological Survey of Pakistan |
| 3 | Landcover | Sentinel-2 images | Land use/Landcover (https://earthexplorer.usgs.gov/, accessed on 10 February 2023) |
| 4 | Soil | Soil map | Food and Agricultural Organization (FAO) website (http://www.fao.org/soils-portal/data-hub/soil-maps-and-databases/en/, accessed on 10 March 2023) |
| 5 | Precipitation | GIOVANNI | (https://giovanni.gsfc.nasa.gov/giovanni/, accessed on 15 March 2023) |

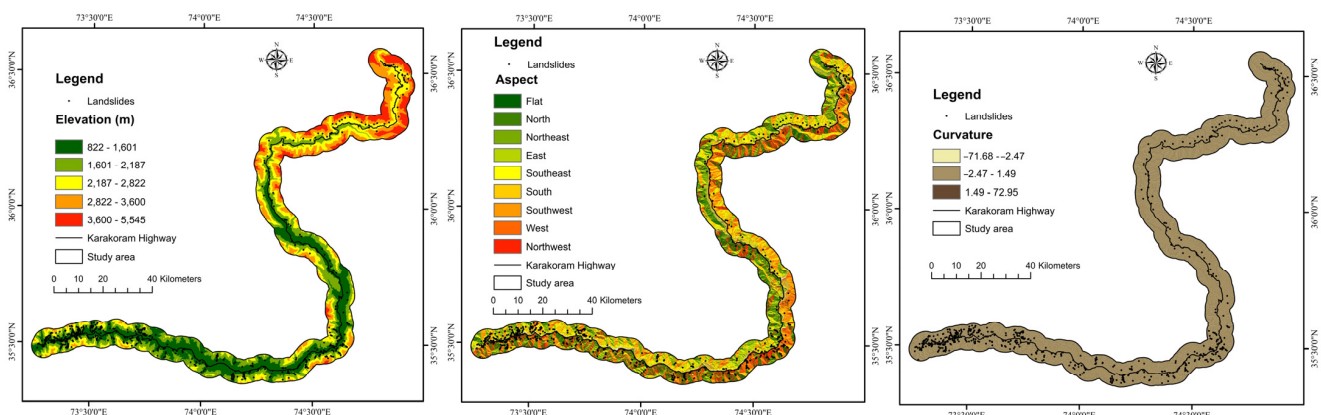

**Figure 5.** *Cont.*

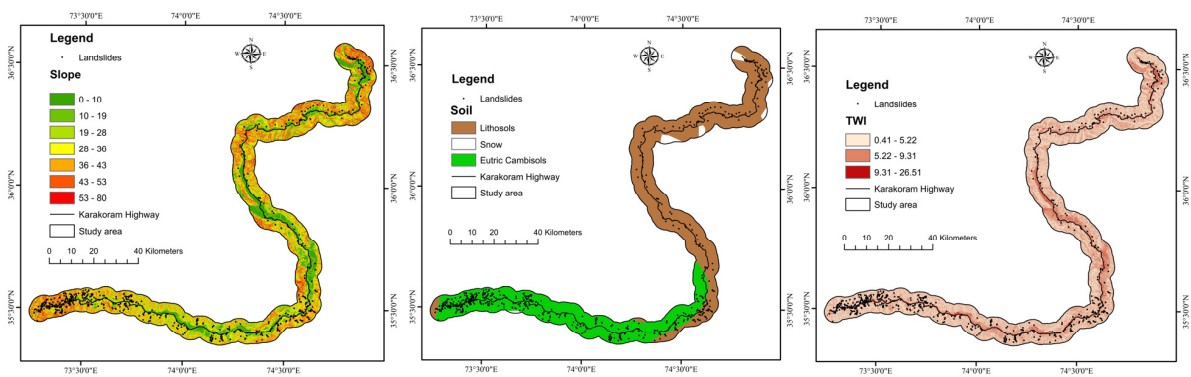

**Figure 5.** LCFs used in the study area.

**Figure 6.** LCFs used in the study area.

## 2.5. Multicollinearity Assessment

In landslide susceptibility prediction, selecting the right variables is a critical step that significantly impacts the model's performance. To improve the accuracy of hazard mapping,

it is essential to identify and choose appropriate variables for inclusion in the model [67]. One of the main challenges in variable selection is dealing with multicollinearity, which may arise due to the improper use of redundant or highly correlated factors [68]. To address the issue of multicollinearity, tolerance and variance inflation factors (VIF) are calculated. These measures help to identify closely and linearly related variables in a regression model, which could potentially lead to a reduction in the model's performance [69]. Through the standard equations for VIF and tolerance, a thorough evaluation of the degree of multicollinearity in the data is conducted. By identifying and removing variables with high levels of multicollinearity, the precision of landslide susceptibility models can be enhanced, and the most significant variables contributing to the prediction of landslide hazards can be identified more accurately.

$$\text{TOL} = 1 - R_j^2 \tag{1}$$

$$\text{VIF} = \frac{1}{1 - R_j^2} \tag{2}$$

where $R_j^2$ represents the regression value of $j$ on various factors. Thus, multicollinearity problems usually happen if the TOL value is <0.10 and the variance inflation factors value is >10 [70].

### 2.6. Machine Learning Models

The modeling process included fitting, identifying, and developing an ML model.

The grid unit was used as the model unit, with a spatial resolution of 12.5 m for both the DEM and RS data, and all evaluation factors are recalculated at this level.

The model includes 15 conditioning factors and a landslide target variable (1 indicating landslide and 0 indicating non-landslide), with each row producing an object.

Each column illustrates an object's characteristic, and it is modified into a two-dimensional matrix for training (70%, 2138 samples) and testing (30%, 917 samples).

The models are constructed using training data, and predictions are made using test data. The landslide vulnerability index maps are produced by combining the prediction values for each model unit in each group. The results of the four models are transferred to a geographic information system. Landslide vulnerability is classified into five categories using Jenks natural breaks [71]: very low, low, moderate, high, and very high. The four models are tested using the ROC curve and the AUROC curve.

### 2.6.1. RF

Random forest (RF) is a well-known homogeneous bagging-based ensemble model developed by Breiman in 2001 [72]. It consists of multiple decision trees, making it an ensemble learning technique that aggregates the outputs of these trees to produce a classification [72–74]. The mechanism of the RF model can be outlined as follows:

I.  Using the bootstrap approach, it creates numerous decision trees to randomly choose fresh sample sets from the initial training dataset with substitution.
II.  At each resampling, a set of features is randomly selected, and the decision trees are built based on this subset of features.
III.  The generated trees are combined into an RF, which is then used to categorize new data.

The RF method exhibits robustness against missing, unbalanced, and multicollinear data and is capable of handling high-dimensional data [75,76]. One of its primary advantages is its resistance to overfitting, even when a large number of random forest trees are grown. Additionally, there is no need to rescale, transform, or modify the data when applying the RF algorithm. In this research, the landslide susceptibility model was developed using the "randomForest" package in R 4.0.2 software [16]. After numerous attempts, the number of trees (ntree) was set to 500, and the mtry parameter was set to 6. To minimize the fluctuation of the model findings and to limit overfitting, RF was conducted using a

10-fold cross-validation technique. The hyperparameters used in the RF model are listed in Table 3.

**Table 3.** List of parameters used in random forest model.

| Parameters | Values |
|:---:|:---:|
| Node size | 14 |
| mtry | 06 |
| ntree | 500 |

### 2.6.2. XGBoost

The XGBoost supervised classification model is built on the gradient tree boosting algorithm [77,78], which is a powerful machine learning model designed by Chen and Guestrin in 2016 [79]. This model creates consecutive decision trees using the estimated residuals or errors from the preceding tree rather than integrating separate trees. This approach allows the algorithm to focus on samples with higher uncertainty, improving its performance. XGBoost offers several advantages, including scalability for various use cases with low computing resource essentials, fast processing speed, efficient handling of sparse data, and smooth integration [80]. The algorithm utilizes a loss function with an additional regularization term to smooth the final learned weights and prevent overfitting [79]. It also employs first- and second-order gradient statistics to optimize the loss function [81]. While XGBoost shares some parameters with other tree-based models, it involves additional hyperparameters to control the overfitting concern, enhance precision, and mitigate forecasting variance [82]. This study develops the landslide susceptibility model using the "XGBoost" package in R 4.0.2 software, which provides powerful capabilities for classification tasks [83]. In this research, three general parameters were chosen for us to alter in the XGBoost algorithm for LSM application: nrounds (the maximum number of boosting repetitions), subsample (the subsample ratio of the training instance), and colsample_bytree (the subsample ratio of columns while constructing each tree). The hyperparameters used in XGBoost model are listed in Table 4. The key points and usability of machine learning models are shown in Table 5.

**Table 4.** List of parameters used in extreme gradient boosting model.

| Parameters | Values |
|:---:|:---:|
| nround | 210 |
| subsample | 1 |
| colsample_bytree | 0.75 |
| max_depth | 6 |
| gamma | 0.01 |
| eta | 0.05 |

**Table 5.** The key points of RF and XGBoost models.

| RF | XGBoost |
|:---:|:---:|
| Bagging ensemble method | Boosting ensemble method |
| Bagging-based algorithm where only a subset of features are selected at random to build a forest or collection of decision trees | Gradient boosting employs gradient descent algorithm to minimize errors in sequential models |
| Reduce risk overfitting | Regularization for avoiding overfitting |
| Maintain precision when a large proportion of data is missing. | Efficient handling of missing data |
| Time-consuming process | Less time-consuming process |

### 2.7. Deep Learning Models

#### 2.7.1. CNN-2D

As a supervised DL approach, the CNN excels in achieving high predictive performance in fields such as image and speech recognition. It accomplishes this by hierarchically composing simple local features into complex models. A typical CNN comprises one or more convolutional layers, fully associated layers, and max pooling layers, enabling it to classify and extract features from high-dimensional data [84]. In the context of landslide susceptibility mapping (LSM), the landslide occurrence potential in each grid cell is influenced by multiple factors. Each grid cell possesses a unique set of characteristic values that illustrate the likelihood of a landslide event. We must initialize the process to apply the CNN for LSM by transforming the 1D input grid cell containing various characteristic attributes into a 2D matrix.

We compared the number of landslide-impacting variables with the number of characteristic values for every variable in this study. The largest of these two integers was then chosen to define the size of the related 2D grid. For example, if the research region has 9 lithological classes and 15 landslide influencing factors, we create $9 \times 9$ matrices for each grid cell. In the matrix, each column vector represents an attribute value, and the element at the corresponding position is assigned the value of 1. In contrast, other elements in the vector are assigned the value of 0. Some of the predictive parameters utilized in the present research for CNN models are provided in Figure 7 and Table 6.

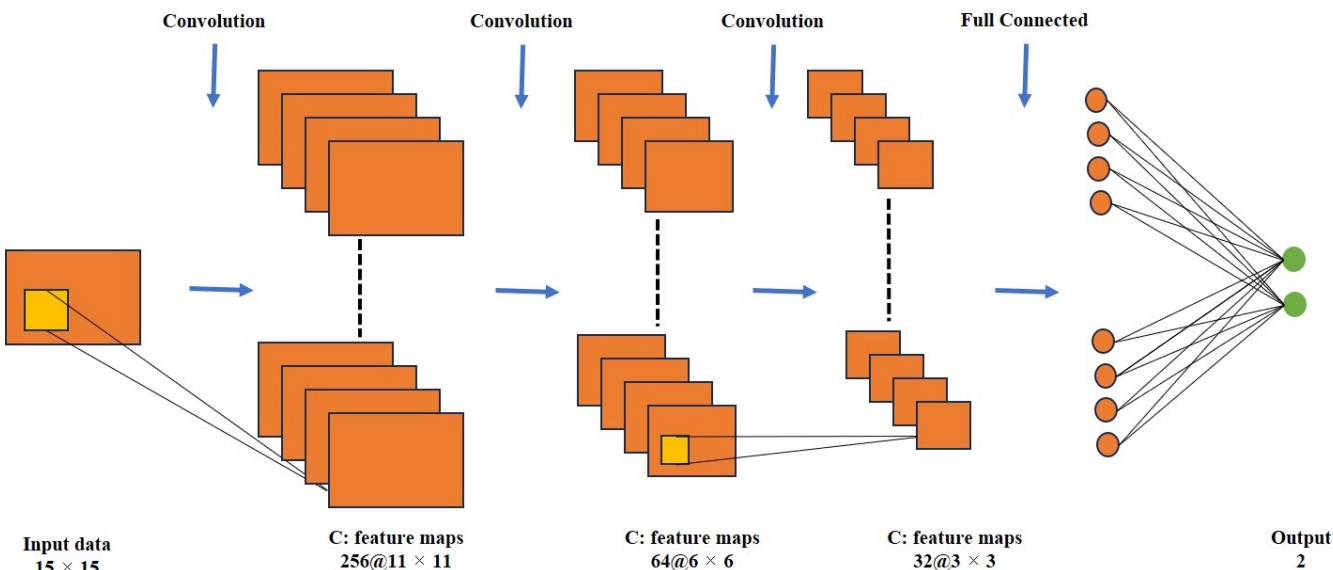

**Figure 7.** The structure of CNN-2D is illustrated in a schematic figure.

**Table 6.** List of parameters used in CNN-2D model.

| Parameters | Values |
|---|---|
| Batch size | 8 |
| Epoch | 250 |
| Dropout | 0.5 |
| Learn rate | 0.002 |
| Activation function | ReLU |
| Optimizer | Adam |

#### 2.7.2. RNN

An RNN (recurrent neural network) has the ability to capture dynamic information in sequential data by creating connections between hidden layer nodes at different time steps. Unlike other neural networks, RNNs can effectively leverage sequential data. In

traditional neural networks, all inputs are treated as self-reliant entities. However, in the RNN technique, each unit is linked to other units in the hidden layer at various time intervals, allowing data to be propagated from one layer to the next in the network [85]. This characteristic of RNNs is achieved through the concept of "loop feedback," where information is shared throughout the RNN. A simple RNN is typically executed using Jordan or Elman network architectures. At time step t, let $x_t$, $y_t$, and $h_t$ represent the input vector, the output vector, and the hidden state vector, respectively. By utilizing these elements, we can acquire:

$$h_t = \sigma(W_h x_t + U_h h_{t-1} + b_h) \tag{3}$$

$$y_t = \sigma(W_y h_t + b_y) \tag{4}$$

where $\sigma(\cdot)$ is the training sample sequence's loss function, $U$ and $W$ are variable matrices, and $b$ is the appropriate bias vector.

RNN is particularly adept at manipulating sequential inputs through its recurrent hidden states. Hence, the accurate visualization of data is crucial in realizing the forecasting capability of RNNs. This portion presents the data visualization method for landslide susceptibility mapping using RNNs, as illustrated in Figure 8. The parameters used in the RNN model are listed in Table 7.

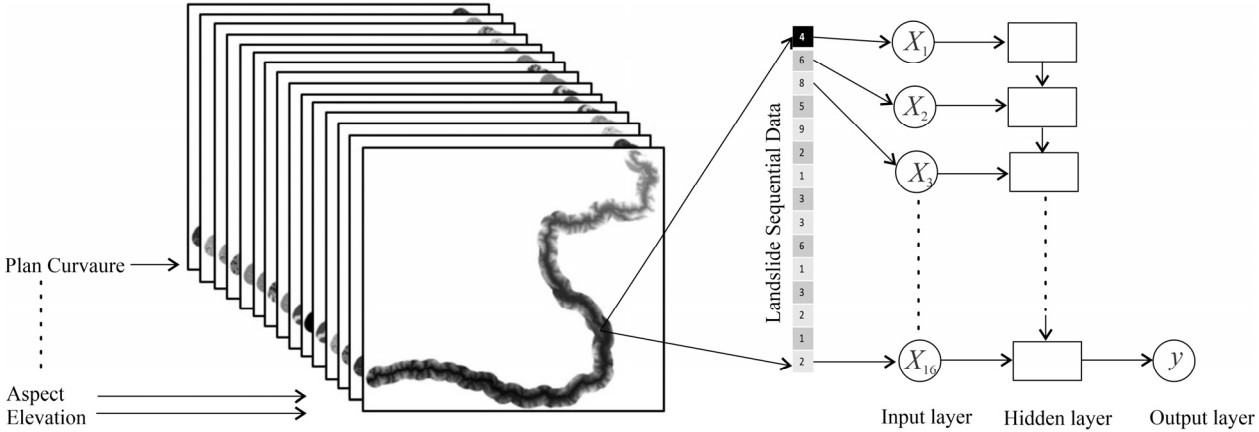

**Figure 8.** Data representation for RNN.

**Table 7.** List of parameters used in RNN model.

| Parameters | Values |
|:---:|:---:|
| Batch size | 64 |
| Epoch | 50 |
| Dropout | 0 |
| Learn rate | 0.001 |
| Optimizer | Adam |

First, each LCF is treated as a single-band image, and all variable categories are assembled. Subsequently, these variable categories are then ordered in a decreasing sequence of relevance. Hence, each pixel can be transformed into a sequential sample based on its importance level. This approach ensures that the most significant variables are fed to the RNN framework first, while the less crucial variables are sent to the model last. Because of the recurrent nature of the RNN, essential data contributing to landslide occurrences are reserved and passed to the next hidden state. This retention of important information is advantageous for the final LSM. By organizing the data representation in this manner, we can effectively leverage the sequential processing capabilities of the RNN to improve the precision of the LSM. The key points and usability of DL models are in Table 8.

**Table 8.** The key points of CNN-2D and RNN models.

|  | CNN 2D | RNN |
|---|---|---|
| Basics | Most popular type of neural networks | Most advanced and complex neural network |
| Structural layout | Structure is based on multiple layers of nodes including one or two conventional layers | Information flows in different direction, which gives it its self-learning feature and memory |
| Spatial recognition | Yes | No |
| Recurrent connection | No | Yes |
| Drawback | Large training data required | Slow and complex training and gradient concern |

## 2.8. InSAR

The InSAR technique has proven to be highly valuable for the early identification of landslides due to its weather independence, wide monitoring coverage, and high accuracy. Among the various InSAR techniques, the small baseline subset (SBAS) is particularly useful for identifying slow-moving deformations with millimeter-level precision by utilizing a stack of SAR interferograms [86]. Additionally, the PS-InSAR and SBAS-InSAR techniques are utilized to evaluate the deformation in susceptible regions as generated by the models. This research collects and evaluates imagery from the Sentinel-1A IW sensor with a temporal resolution of 12 days.

### 2.8.1. SBAS-InSAR Processing

The SBAS-InSAR technology is used in this part to verify the LSM along the KKH. The basic data analysis chart, shown in Figure 3, incorporates the preprocessing of data, interferometric creation, phase unwrapping, refinement and re-flattening assessment, and displacement estimations.

The computation of time and spatial baselines between all Sentinel-1 picture pairs is part of data preparation. Following clipping and registration, the DEM data are utilized to finish image authorization, and the proportional conjunction that meets a particular threshold is chosen to generate a differential SAR interferogram set [87]. This investigation used a 30 m resolution SRTM-DEM to construct interferograms. The super primary image utilized comes from the images acquired on 15 July 2022, and 720 interferometric image pairings were created.

The key phase of SBAS-InSAR manipulation is an inversion, and the displacement computation is highly dependent on the investigation of inversion findings. The displacement rate and residual topography are estimated in the first inversion, and the input interferogram is optimized in the second unwrapping [88]. The second inversion expands upon the first by employing low-pass and high-pass filtering to calculate and eliminate the atmospheric phase, allowing for more accurate final displacement estimates and, ultimately, geocoding to determine the displacement rate dispersion in the research region. To avoid the effects of unwrapping inaccuracies, the line-of-sight displacement velocity was computed using a coherence threshold of 0.3 for SBAS-InSAR analysis [89].

### 2.8.2. PS-InSAR Processing

The PS-InSAR process analyzes the uniformity of the phase and amplitude using multitemporal SAR images wrapped around the same area, which determines the pixels that are not as caused by spatiotemporal decorrelation and then defines specific displacement information on each component of the phase, which must be conjunctly assessed and modeled to remove discrepancies [90,91].

The data preparation analysis steps (Figure 3) for importing SLC data with accurate paths comprise the following:

Acquiring imagery: Images with the same rotations are obtained, and both slave and master images are selected. The master images of the research area are obtained first, followed by the selection of slave images that overlap the same region.

Coregistration and examination: A specific area of interest is coregistered and evaluated. Various measures such as atmospheric phase screen (APS), track errors, and other factors are corrected and measured.

Phase constancy assessment: The phase constancy of the acquired data is evaluated. The pixels are projected to exhibit similar amplitudes and reduced phase distributions for these acquisitions. Absolute amplitude levels are not a significant concern in terms of manipulation disturbances.

Amplitude stability index (ASI): The ASI is used to choose persistent scatterers (PS) in the SARPROZ software (2023) procedure. PS points with ASI values greater than 0.7 are selected. This constraint parameter ensures that only a limited number of PS points are considered, which is necessary for accurate atmospheric phase screen computation.

Reference network and linear model: A reference network is built by linking PS points using Delaunay triangulation. The extracted linear model is removed, including residual height and linear deformation velocities. The APS is analyzed using an inverse network from the phase residual, and a single point of reference is defined to estimate the object's velocity.

Multi-image sparse point (MISP) processing: Second-order PS points are chosen using the criteria of ASI > 0.6 in this step. Thicker PS points are obtained at this phase. To eliminate APS, identical parameters and reference points used for APS estimates are applied.

Geocoding and visualization: Google Earth is used to geocode and map the PS points. The landslide susceptibility map only includes PS points with a coherence of 0.60, indicating their reliability [92].

### 3. Results

*3.1. Multicollinearity Analysis*

This research assumed that there are no significant linear associations among the landslide causative variables that can negatively impact the susceptibility models. A multicollinearity analysis is conducted on the 15 landslide conditioning variables, and the outcomes are presented in Table 9. The rainfall variable has the highest VIF score of 4.892, while the lowest VIF score of 1.017 is observed for Aspect. The TOL values ranged from 0.204 to 0.982.

**Table 9.** Multicollinearity assessment of the LCFs.

| S.No. | Variables | Collinearity Statistics | |
|---|---|---|---|
| | | TOL | VIF |
| 1 | Aspect | 0.982 | 1.017 |
| 2 | Landcover | 0.826 | 1.209 |
| 3 | Rainfall | 0.204 | 4.892 |
| 4 | Geology | 0.549 | 1.821 |
| 5 | TWI | 0.655 | 1.524 |
| 6 | Soil | 0.284 | 3.509 |
| 7 | Slope | 0.531 | 1.881 |
| 8 | Distance to Fault | 0.979 | 1.021 |
| 9 | Distance to Road | 0.783 | 1.275 |
| 10 | Distance to River | 0.641 | 1.432 |
| 11 | Roughness | 0.637 | 1.542 |
| 12 | Elevation | 0.375 | 2.662 |
| 13 | Profile Curvature | 0.800 | 1.249 |
| 14 | Plan Curvature | 0.922 | 1.084 |
| 15 | Curvature | 0.779 | 1.282 |

### 3.2. Landslide Susceptibility Mapping

The landslide susceptibility maps are generated using deep learning and machine learning models: CNN 2D, RNN, XGBoost, and RF (Figure 9). The experiment showed that the CNN 2D model had the best performance among the four models. The LSMs were classified into five categories using the Jenks natural break [71] technique in ArcGIS.

The precision of the landslide susceptibility maps was assessed using the confusion matrix proposed by [32]. Table 10 shows the performance of the CNN 2D, RNN, XGBoost, and RF models during the training stage. The outcomes revealed that the CNN 2D model has a high precision rate of 0.836 in the study region. Further validation was performed using the ROC (receiver operating characteristic) method [93], which plots "sensitivity" vs. "specificity" for different cut-off estimates. However, to understand the model's performance, the ROC's AUC was used [94]. The AUC results for the CNN 2D, RNN, XGBoost, and RF models are 82.56%, 79.43%, 76.04%, and 75.37%, respectively (Figure 10).

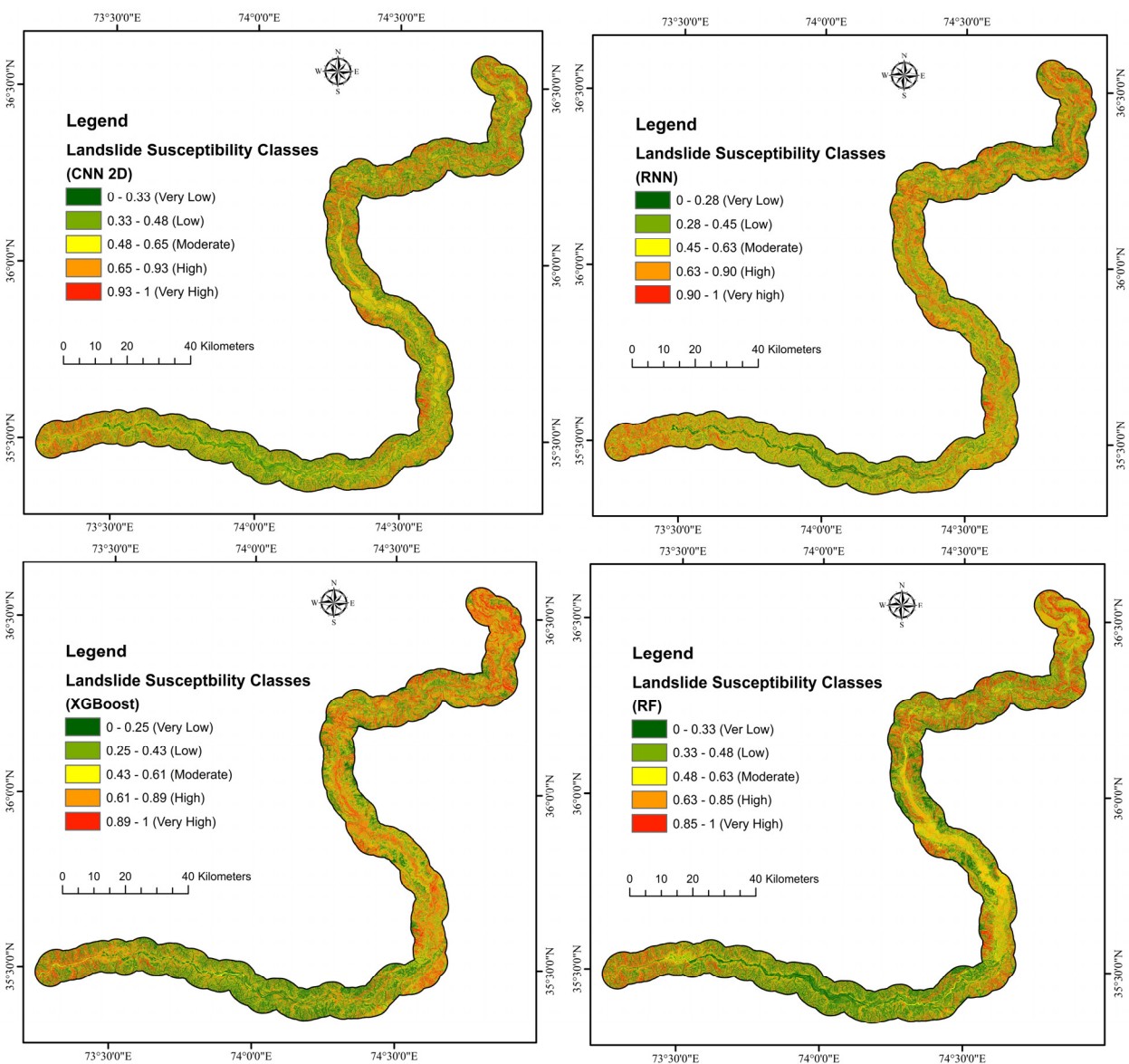

**Figure 9.** LSM using CNN 2D, RNN, XGBoost, and RF models.

**Table 10.** Confusion matrix of CNN 2D, RNN, XGBoost, and RF models.

| Models | Observation | Predicted | | Precision |
|---|---|---|---|---|
| | | **No** | **Yes** | |
| CNN 2D | No | 46 | 53 | 83.61 |
| | Yes | 127 | 872 | |
| RNN | No | 38 | 49 | 83.24 |
| | Yes | 135 | 876 | |
| Extreme Gradient Boosting | No | 41 | 55 | 83.01 |
| | Yes | 132 | 870 | |
| Random Forest | No | 36 | 58 | 82.24 |
| | Yes | 137 | 876 | |

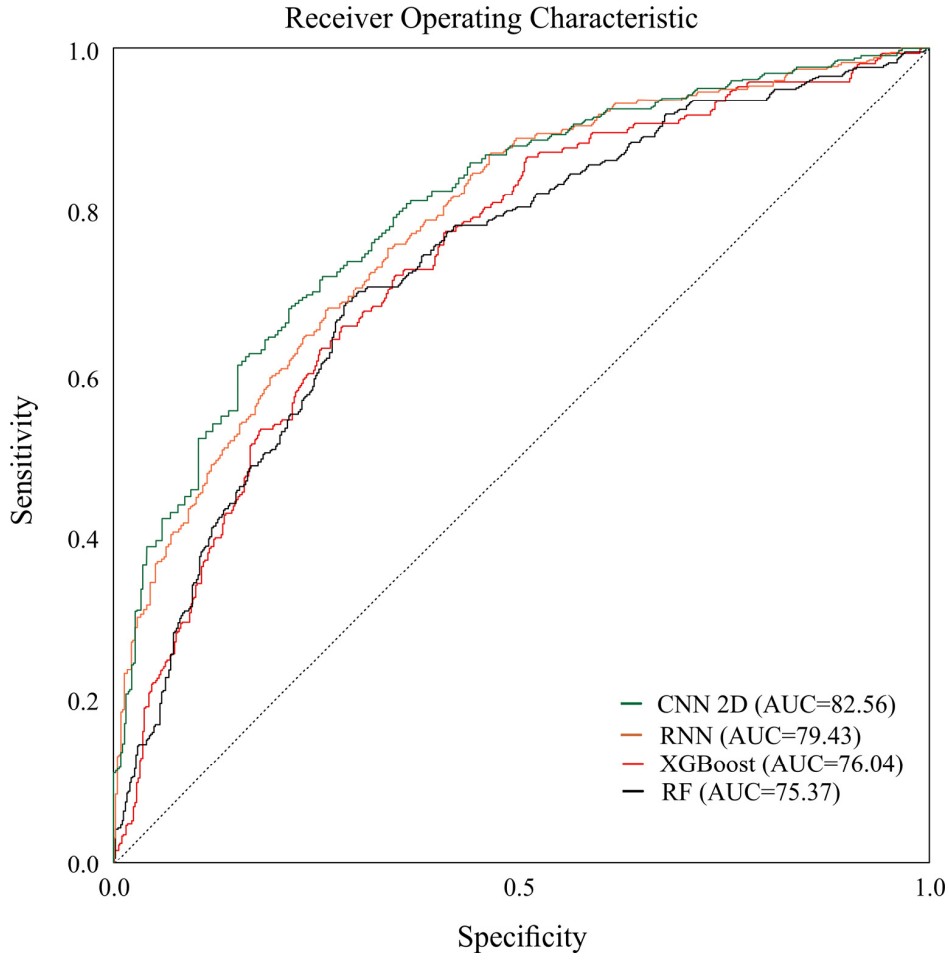

**Figure 10.** ROC plots of CNN 2D, RNN, XGBoost, and RF models.

### 3.3. Landslide Mapping Based on Deformation Velocity

A thorough landslide inventory was organized for the identification and analysis of landslides along KKH. This inventory was built by combining data from several sources, including displacement values acquired from both descending and ascending data gathered from PS-InSAR and SBAS-InSAR observation.

In this study, the InSAR analysis successfully detected the majority of previously mapped landslides. Moreover, based on the PS and SBAS data, several new landslides with significant deformation velocity were identified, along with the boundaries, which

were calibrated through fieldwork. The PS and SBAS techniques were applied to derive displacement rates along the time series in a one-dimensional LOS direction using a set of highly coherent interferograms with small spatial and temporal baselines [95,96].

### 3.3.1. PS-InSAR Results

Surface deformation on the Earth's surface was calculated using a temporal coherence threshold of 0.7 for PS-InSAR analysis. A total of 324,747 PS/DS target points were obtained, representing the LOS deformation values ranging from −92.37 to 72.28 mm/year. These values were converted into Vslope using the transformation formula (5), resulting in a total of 212,373 points. The maximum slope displacement velocity was determined, with an appropriate threshold set between 0 and −20 mm/year (Figure 11). It was ascertained that barren terrain has a higher concentration of PS points than forested regions.

$$Vslope = \frac{\text{VLOS}}{\cos\varnothing} \tag{5}$$

where VLOS is displacement and $\varnothing$ is the incident angle.

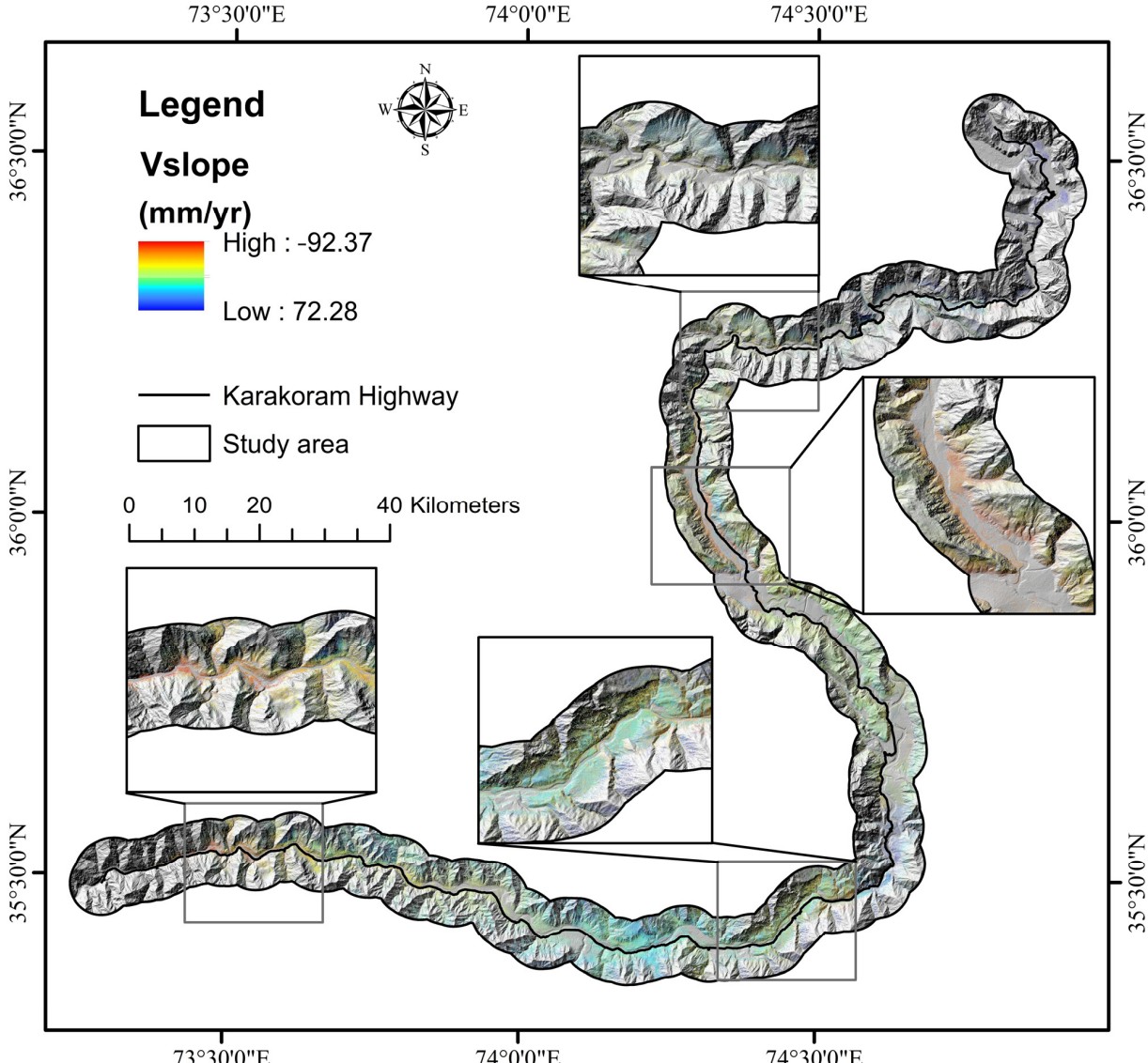

**Figure 11.** Displacement velocity along landslides and slope measured by using PS-InSAR applying the ascending and descending orbit data.

As a result, a total of 15 potential landslides were identified and detected based on the PS-InSAR-identified displacement velocity. Among the 15 landslides detected using PS-InSAR, 10 were exclusively identified using ascending Sentinel-1 datasets, while 5 were specifically detected using descending Sentinel-1 datasets (Figure 12). This observation demonstrates that the combination of descending and ascending datasets can overcome the constraints of acquiring data from a single scanning posture, improving the landslide detection process.

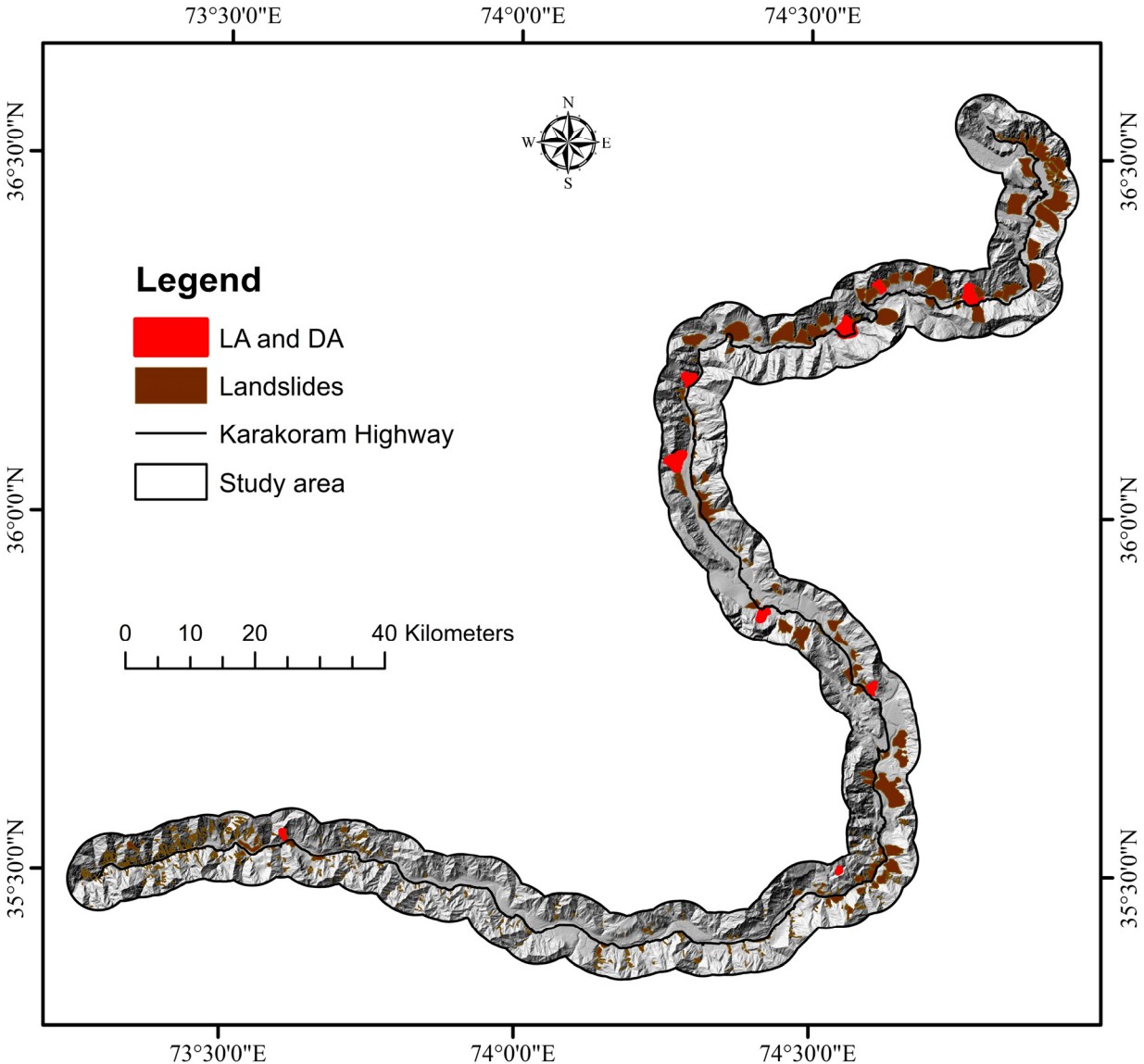

**Figure 12.** Landslide distribution identified through multi-track Sentinel-1 datasets based on PS-InSAR. LA and LD: the landslide identified from ascending and descending Sentinel-1 datasets.

In the identification and detection of landslides, the PS-InSAR identifies displacement data and characteristics from images and field photographs, which are found valuable when used in integration. Most of these landslides are concealed by PS-InSAR-detected coherent targets (Figure 13). However, some landslides may not strictly meet the criteria of higher deformation velocity to be identified as active landslides, as they might experience lower rates of deformation due to steeper slopes or being in an actual inactive state.

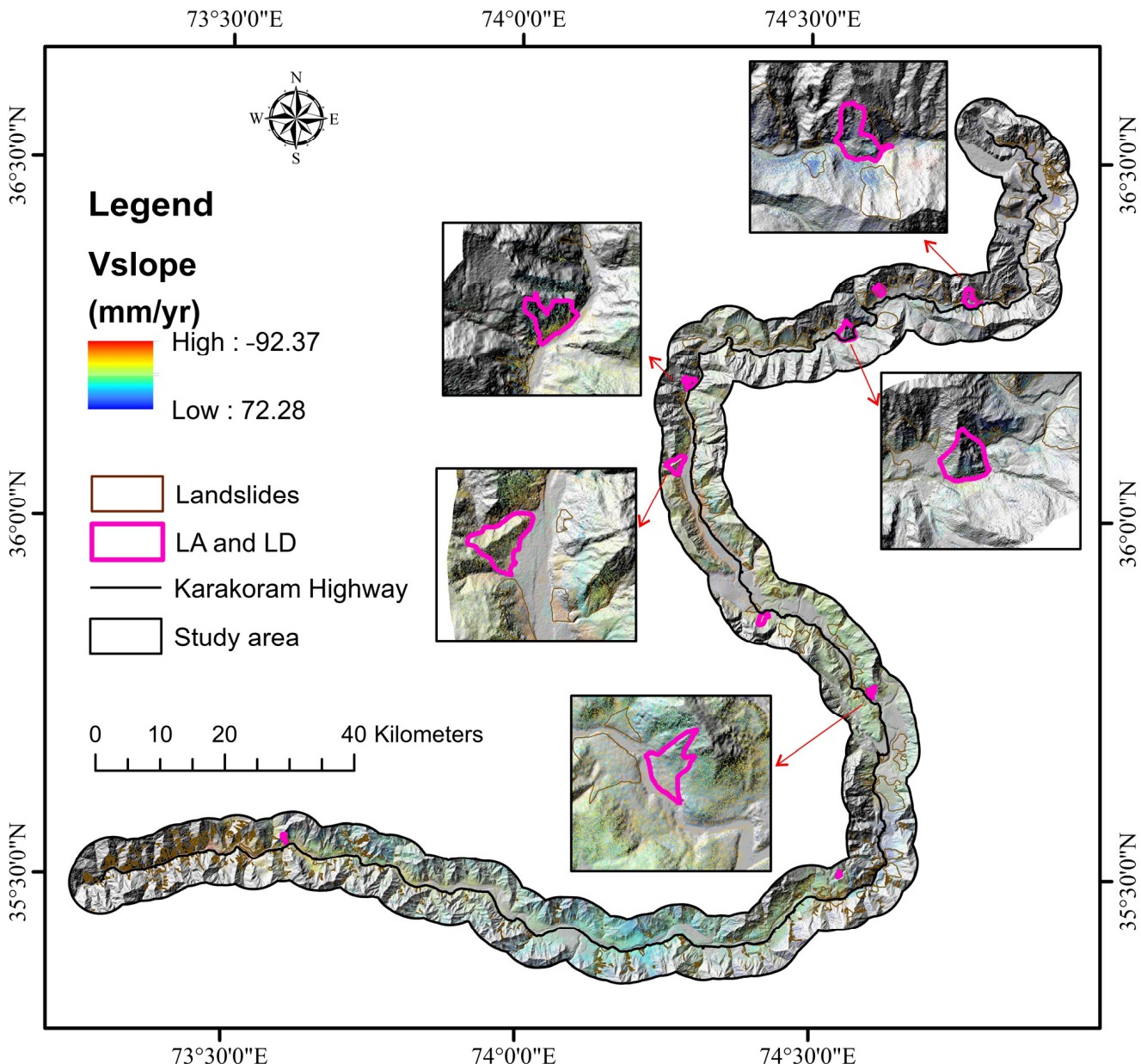

**Figure 13.** PS-InSAR displacement observation and optical image assessment of identified and interpreted landslides.

### 3.3.2. SBAS-InSAR Results

In the SBAS-InSAR processing, the LOS displacement velocity (VLos) was determined using a coherent threshold of 0.3. The slope orientation velocity (Vslope) was then derived from the satellite LOS data, showing only unidirectional displacement. Since landslides and Earth's surface deformations mostly happen over steep land, Vslope is an essential constituent used to forecast landslide evolution. The SBAS-InSAR results indicate that the displacement velocity along the LOS ranged from −81.89 to 75.40 mm/year (as depicted in Figure 14).

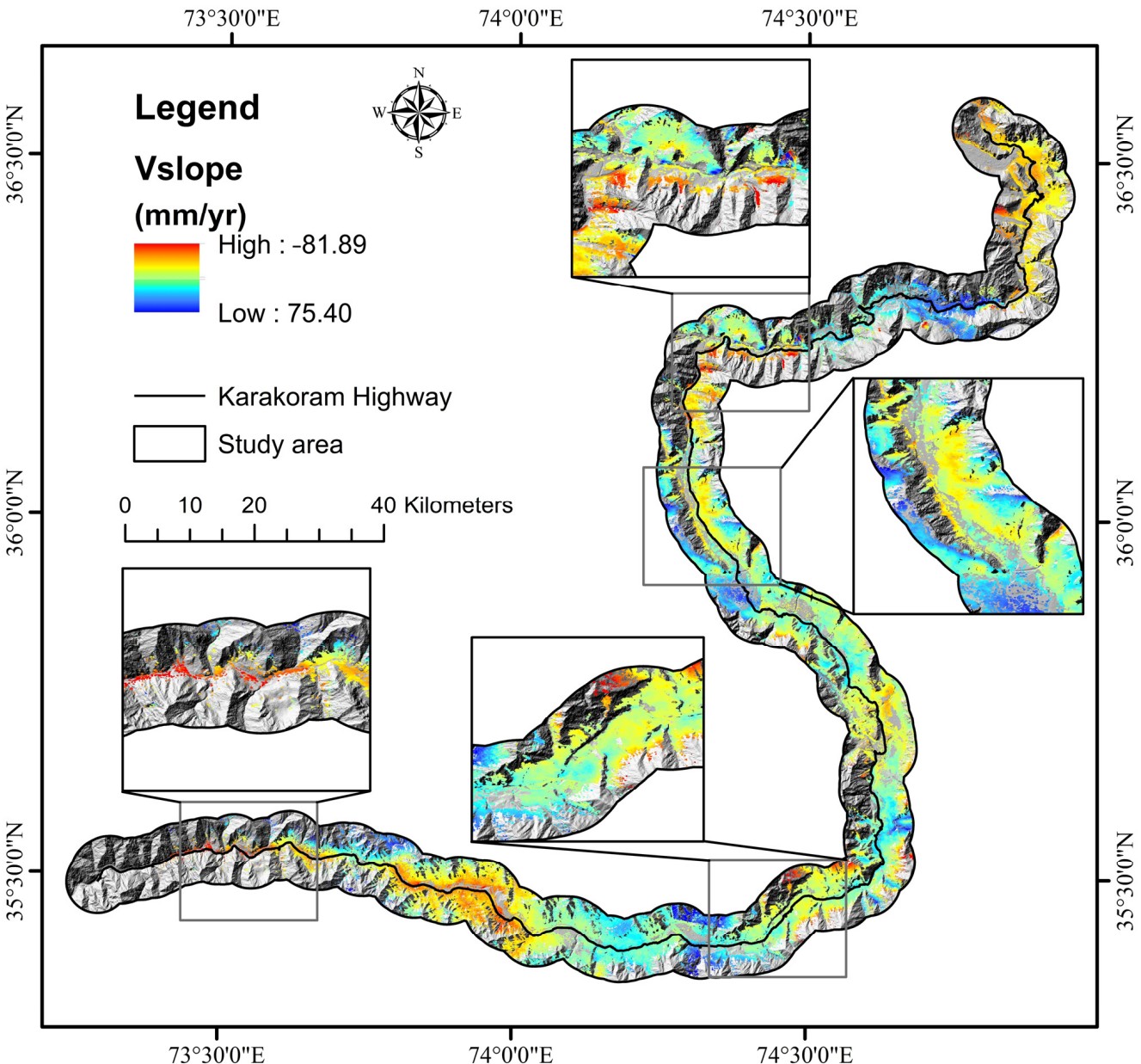

**Figure 14.** Displacement velocity along landslides and slope measured by using SBAS-InSAR applying the ascending and descending orbit data.

The preliminary delineation of landslide boundaries was conducted by combining the displacement velocity along the slope from both descending and ascending datasets with visual analysis of optical RS images and field assessments. This involves referring to areas with relatively high deformation velocity, topographic characteristics obtained from the digital elevation model (DEM), and features observed in the optical images.

As a result of the SBAS-InSAR analysis, a total of 9 potential landslides were detected and identified. Additionally, 547 landslides were represented through a combination of information from the literature [12,60,66] and field observations (Figure 15). Among the 9 SBAS-InSAR-identified landslides, the ascending Sentinel-1 dataset identified 6, and 3 were specifically identified using the descending Sentinel-1 dataset.

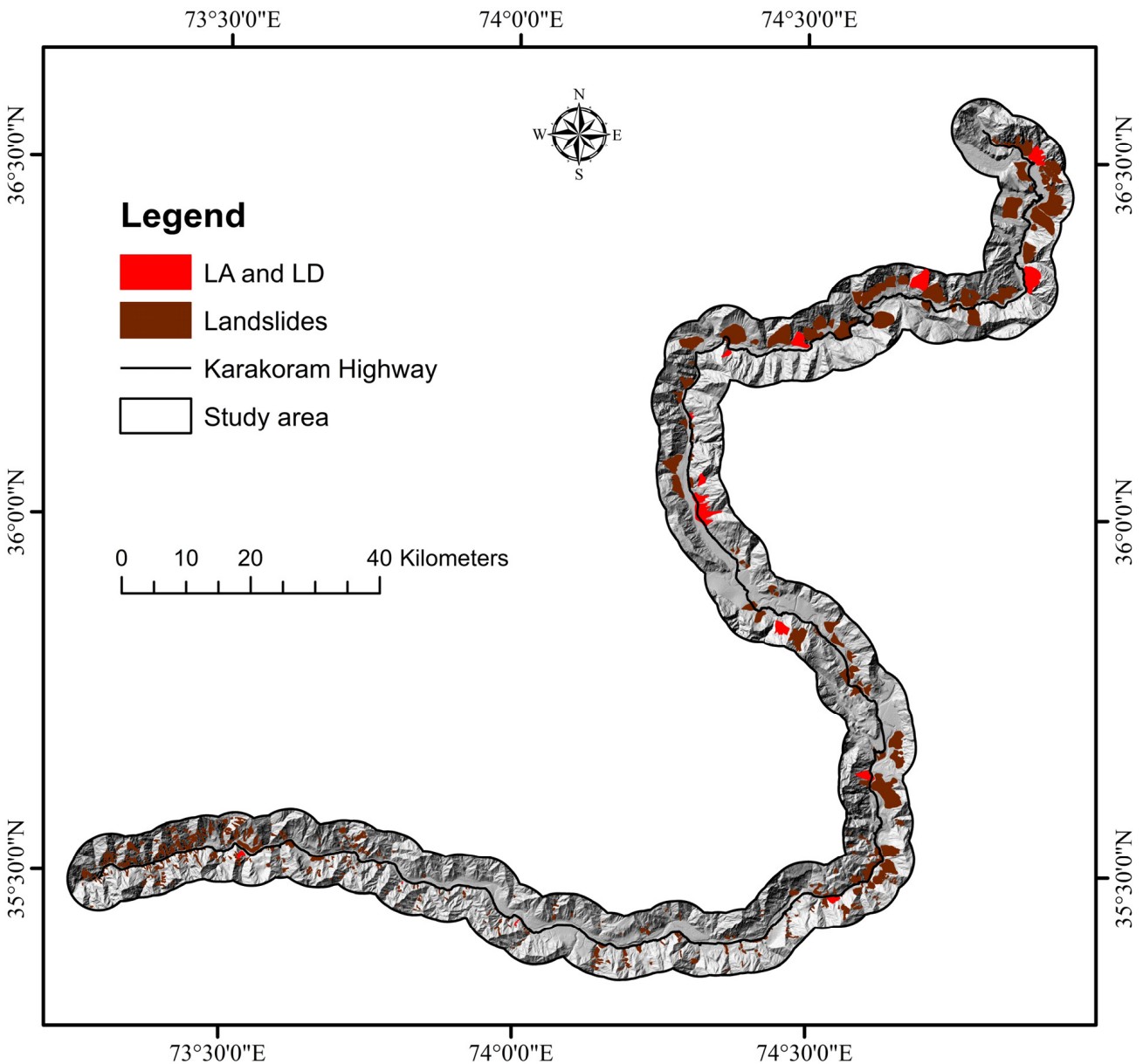

**Figure 15.** Landslide distribution identified through multi-track Sentinel-1 datasets based on SBAS-InSAR. LA and LD: the landslide detected by using ascending and descending Sentinel-1 dataset.

Figure 15 provides comparative examples of identified landslides, most of which are wrapped by SBAS-InSAR-identified coherent targets. The analysis of the landslides revealed that approximately 90% of them are associated with Quaternary deposits, the Hunza plutonic unit, southern Karakorum Metamorphic Complex, Permian massive limestone, and Chilas Complex formations. Their delineation was achieved through field investigations, analysis of optical RS images, and references to the existing literature (Figure 16).

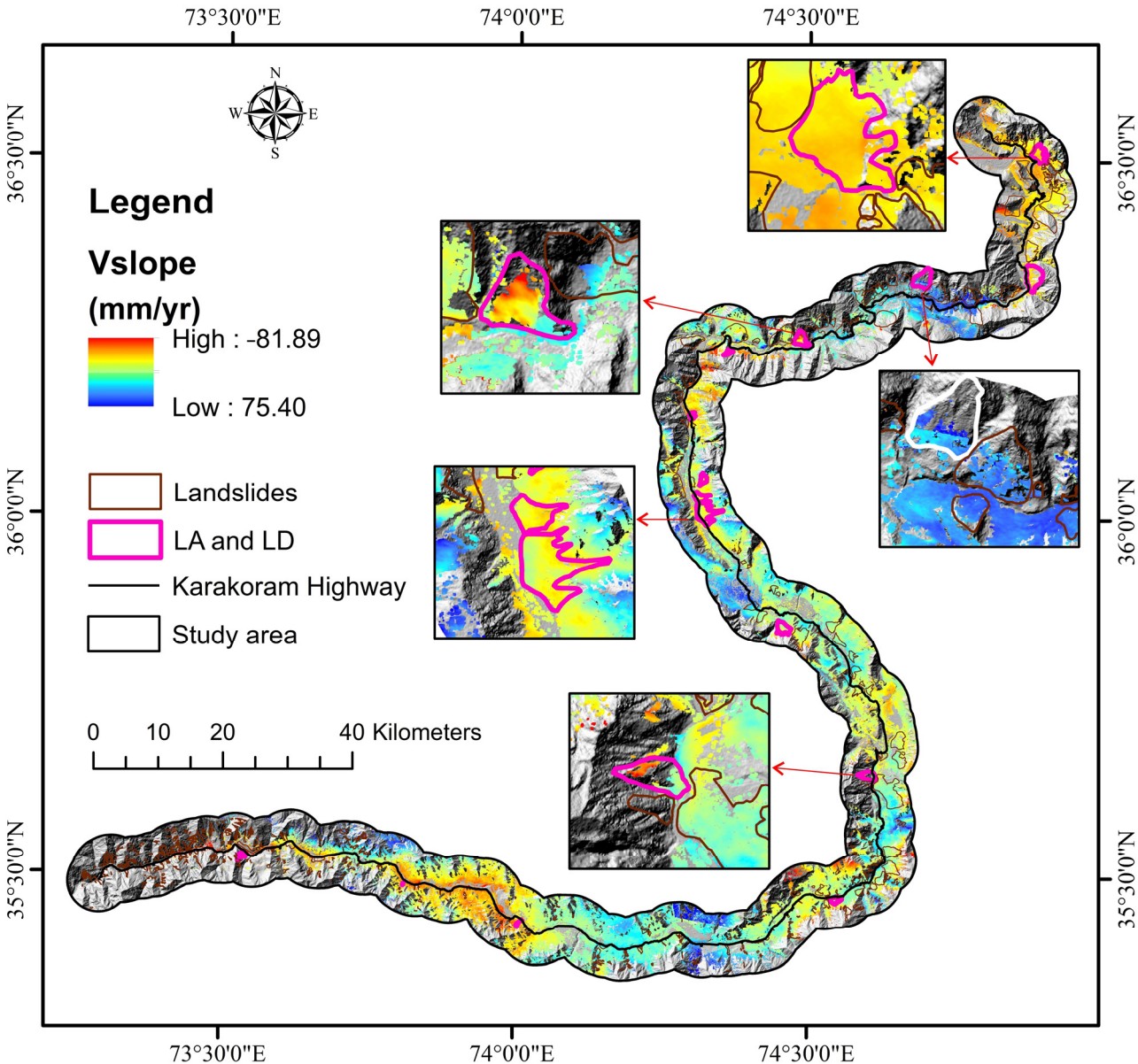

**Figure 16.** SBAS-InSAR displacement observation and the optical image assessment of identified and interpreted landslides.

The majority of KKH's landscape is barren, with nearly 90 percent of the landslides occurring in non-vegetated zones. While SBAS-InSAR and PS-InSAR methods may have limitations in vegetation-covered areas, they still apply to more than 60% of the land devoid of vegetation. This suggests that vegetation plays a critical role in controlling slope stability in the region, consistent with previous research findings [60,97,98].

## 4. Discussion

The current study utilized RS techniques, such as optical RS and InSAR, for risk assessment and landslide mapping along the KKH [63,99,100]. This study took benefit of the multi-azimuth interpretation provided by the descending and ascending Sentinel-1 dataset, allowing for more extensive monitoring of surface displacement. The PS-InSAR and SBAS-InSAR techniques effectively captured regions with high deformation rates in most areas along the KKH. Additionally, the comprehensive landslide inventory presented in this study includes the latest landslides, ensuring the database is up to date and its valuable information. By processing Sentinel-1 data from June 2021 to June 2023, utilizing the InSAR

technique, 24 new prospective landslides were identified, and some existing landslides were redefined. This updated landslide inventory was then utilized to create a landslide susceptibility model, which investigated the link between landslide occurrences and the causal variables. By combining the findings from PS-InSAR, SBAS-InSAR, and field investigations, the inventory was updated with landslides that have the potential for future failure and pose risks for the region, contributing to improved landslide susceptibility mapping.

The selected landslide influencing factors were used to construct CNN 2D and RNN architectures for comparison with the XGBoost and RF methods. The LSMs were validated and compared based on the AUROC curve and accuracy. CNN is known for its ability to efficiently obtain spatial data using weight sharing and local connections, making it a promising method for landslide modeling [101]. Earlier research has shown that combining CNNs with additional statistical approaches can produce better accuracy in landslide susceptibility modeling than using CNNs alone [102]. According to [103], CNN models are an improved tool for landslide modeling due to their substantial outcomes and higher accuracy rate in spatial landslide forecasting. The outcomes also reveal that both DL and traditional ML algorithms give excellent precision in a variety of sectors, such as landslide assessment and earth science studies throughout the world [104], which is in line with our findings, which showed that the ROC for the four models varies from 82.56 to 75.37%.

In the subsequent experiments, the proposed CNN and RNN models demonstrated enhanced predictive capability compared to the popular XGBoost and RF classifiers. Specifically, CNN-2D attained the highest AUC value of 0.825 on the validation set, indicating its effectiveness in improving prediction performance and its potential as a potential approach for future research. Various statistical and machine learning approaches have been compared and applied for landslide spatial forecast in areas, including AHP and Scoops 3D [105], frequency ratio (FR) and weight of evidence [106], the weighted overlay technique and AHP [9], random forest [63], support vector classification (SVC) [107], and XGBoost [100], but DL techniques, such as CNNs, provide powerful improvements by automatically exploring representations from raw data, making them valuable in various fields, including landslide susceptibility assessments. The experimental findings highlighted that CNN-2D outperformed the traditional DL approach of RNNs and the classical XGBoost and RF ML techniques. Furthermore, the suggested data representation techniques offer an innovative approach to handling raw landslide data. By exploiting the power of DL methods and combining them with other approaches, there is enormous potential to advance landslide susceptibility analysis in the future. The proposed 2D CNN structure includes convolutional max pooling layers and a dropout layer. Overfitting is a common issue when utilizing a 2D CNN in LSM. To address overfitting, each convolution layer is subsequently followed by a dropout layer, which temporarily discards NN units during the training process of the CNN based on a certain probability. This helps to improve classification accuracies and enhance the model's generalization capability.

Different LCFs influence landslide triggers and relate to each other, making the selection of appropriate variables crucial for building an accurate landslide susceptibility model. The aim is to construct models with reduced noise and greater forecast ability. Before analyzing landslide susceptibility, it is essential to evaluate the forecasting potential of each contributing factor. To attain this, efforts are made to select the most relevant and impactful factors. Multicollinearity analysis is employed to evaluate correlations between the LCFs. In this study, 15 landslide conditioning variables were chosen as independent factors for evaluating landslide susceptibility, and the results are presented in Table 3. The variance inflation factor (VIF) was used to test the multicollinearity between these factors. Among the selected factors, rainfall had the highest VIF score of 4.892, while aspect had the lowest VIF score of 1.017. The tolerance (TOL) values ranged from 0.204 to 0.982. The outcomes revealed that there is no significant multicollinearity among the chosen variables, allowing all variables to be integrated into the models. It is worth noting that landslides can still occur in areas with significant vegetation due to rainfall and other external forces.

Despite the beneficial effect of the selected factors in evaluating landslide susceptibility, the current research could have been more effective if certain factors were adhered to. One major factor is data availability. This research relied on limited data diversity with a focus on historical landslide data, which limited the comprehensive analysis [60]. Another factor is that the input data resolution remains unpredictable during the data preparation phase, which has been a prevalent issue in past investigations [108,109]. Terrain condition factors were derived from a 12.5 m resolution DEM, while variables related to geological conditions were based on a 1:500,000 scale geological map. All factor layers were resampled at a 12.5 m resolution in ArcGIS 10.8 software to ensure data availability and computational convenience. The analysis of model performance in this study indicates that resampling processing was feasible. Secondly, because of restricted data availability, we examined several types of landslides with varying triggering conditions throughout a given time. While some investigators have previously explored this approach, a separate investigation of distinct types of landslides is more in line with the practical and current state factors [110,111].

## 5. Conclusions

The PS-InSAR and SBAS-InSAR techniques and multi-track ascending and descending Sentinel-1 SAR datasets were used to measure surface displacement velocity along the KKH. An updated and comprehensive landslide inventory was created by combining field surveys, image analysis, and a literature evaluation, identifying 571 landslides, including 24 newly detected active landslides and 547 landslides from previous records. To predict landslide susceptibility along KKH, two well-known deep learning (CNN-2D and RNN) and machine learning (XGBoost and RF) algorithms were utilized and compared. The CNN-2D algorithm demonstrated superior performance with an AUC of 82.56, outperforming RNN, XGBoost, and RF in terms of AUC, ROC, predictive power, and accuracy. The landslide susceptibility maps generated by these models can serve as valuable tools for decision-makers, land use planners, and various non-governmental and governmental organizations involved in resource and disaster management, infrastructure development, and human activity in the study area. In the future, studies can explore improved deep learning and machine learning architectures for landslide susceptibility mapping to improve accuracy and predictive capabilities utilizing this research as a baseline.

The current study is limited because of the absence of geotechnical and geophysical data. It is suggested that the developed dataset be used in future studies to improve algorithm prediction potency, create more precise LSM, and discover correlations between landslide incidence and these new geo-environmental variables. It is also advised to combine DL algorithms with metaheuristic techniques to optimize model parameters and boost algorithm prediction capabilities.

**Author Contributions:** Conceptualization, M.A.H.; Methodology, M.A.H.; Software, M.A.H.; Validation, Y.Z. (Yulong Zhou); Formal analysis, Y.Z. (Yulong Zhou); Investigation, Y.Z. (Ying Zheng); Resources, Y.Z. (Ying Zheng); Writing—original draft, M.A.H.; Writing—review & editing, H.D.; Visualization, H.D.; Supervision, Z.C.; Project administration, Z.C.; Funding acquisition, Z.C. All authors have read and agreed to the published version of the manuscript.

**Funding:** This research was funded by National Natural Science Foundation of China, grant number No. 41871305; National key R & D program of China, grant number No.2017YFC0602204; Fundamental Research Funds for the Central Universities, China University of Geosciences (Wuhan), grant number No. CUGQY1945; Opening Fund of Key Laboratory of Geological Survey and Evaluation of Ministry of Education; and Fundamental Research Funds for the Central Universities, grant number No. GLAB2019ZR02.

**Data Availability Statement:** The data presented in the study are available upon request from the first and corresponding authors. The data are not publicly available due to the thesis that is being prepared using these data.

**Conflicts of Interest:** The authors declare no conflict of interest.

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
