# Peer review of "Deep Learning and Machine Learning Models for Landslide Susceptibility Mapping with Remote Sensing Data"

_remotesensing, doi:10.3390/rs15194703_

Round 1

Reviewer 1 Report

Congratulations to the authors for the work. It is a valuable landslide work for your region. Below are a few minor revision requests.

-In the abstract, you can briefly mention the results numerically.

-Line 158: correct km2 spelling as km2

- Figure 1 ; Region names do not appear on the first map. It can be more legible if the colors are changed. In addition, it is difficult to detect due to the use of black color. It should be rearranged in order to understand which map the given scale belongs to. It seems the legend is the same for both maps, but the scale is different for both.

- Figure 2: You can remove the legend text and give only the lithology title. You can apply this for all other maps. (optional)

- Line 192: sentinel 2 data has been obsolete from the USGS for a while. It would be more appropriate if you could change it to Copernicus.

-Titles 2.3, .2.4, 2.5 are same

- Line 215: You can write 7 instead of 07.

- Figures 5 and 6 both show the criteria. Instead of separating the figures, a title like "Figure 5 cont." can be used.

Reviewer 3 Report

Authors took Karakoram Highway as a case study to do the landslide susceptibility mapping with remote sensing datasets. Deep learning and machine learning models were also used to do this work. Generally, SBAS-InSAR and PS-InSAR technology were also applied to do this work. Finally, the AUC and ROC were used. Generally, LSM is a traditional study field. This study need to add more details before considering its publication.

LINE 14: "a threat to the connectivity.." what's the meaning of connectivity? is there any content missing. 

Line 20: 'based on existing literature', what's the existing litearture, suggest not use this description in abstract part, you can give more detailed description. For instance, based on yearbooks, government reports, etc.

Line 38: 'minor to major' might not be appropriate. Suggest authors to check entire english writing, suggest ask native english speakers to modfiy your english writing.

Line 56: when you first mention KKH, please give full name.

Line 72: in this paragraph, you should give the detailed content of existing LSM methods, give more background description. 

Line 80: traditional methods were missing. At least PCA method, wavelet analysis, and other methods should mention. Some references should be added.

Landslide Susceptibility Mapping Using Ant Colony Optimization Strategy and Deep Belief Network in Jiuzhaigou Region DOI:10.1109/JSTARS.2021.3122825

Line 116: there had some problems in logicality. You should give the background description of methods, data sources. Please rearrange the introduction part.

Section 2.3: The content of how to identify images of past geological disasters is very important and needs further explanation. In addition, the occurrence time is different, and whether the impact of the landslide that occurred at that time or recently can be found is very important. The recommended list is shown in the appendix.

Section 4:

The discussion section is very important and cannot be put together. It is recommended to divide it into several parts. The first part is validation, verifying your landslide identification results and verifying your prediction results. The second part is a comparison with other methods. How is your accuracy? And why do some deep learning or machine learning models achieve better results? What is the reason for this? The third part presents the shortcomings of your research. Are the evaluation indicators you have selected complete? Why are some indicators not yet considered or what elements need to be considered in the future?

Reviewer 4 Report

This study analyzes the susceptibility to landslides along the Karakoram Highway (KKH), a strategically significant route connecting various regions in Asia, which faces high risks due to its severe geological conditions. Utilizing SBAS-InSAR and PS-InSAR technologies, the research identifies and categorizes 571 landslides from June 2021 to June 2023, offering an updated inventory. To understand the relationship between landslide occurrences and their causal factors, the author uses deep learning and machine learning models like CNN 2D, RNN, Random Forest, and XGBoost to model landslide susceptibility based on fifteen identified causative factors. The effectiveness of these models was assessed using the ROC-AUC method, with the CNN 2D model proving to be the most effective in creating a Landslide Susceptibility Map (LSM) for the KKH. Although the paper presents some innovative approaches, certain limitations persist. The following comments and suggestions are provided for the authors to improve their work in a revised version:

1.     Page 2, Line 84: “explainable AI (XAI) model…” Please consider rewording this sentence or changing the reference. The XAI aspect of the models in the cited reference comes from applying SHAP values for interpretation rather than from any intrinsic property of the CNN and SVM models themselves. The authors are encouraged to explore more suitable references (e.g., Youssef et al., 2023, https://doi.org/10.1038/s43247-023-00806-5)

2.     Page 2, Lines 86-90: “Although the Artificial Neural Network (ANN) is a widely used algorithm, it may have limitations in predictive performance, particularly when dealing with testing data beyond the range of the training dataset. On the other hand, Support Vector Machine (SVM) models offer four kernels that help mitigate bias by selecting the most efficient kernel during the modeling process.” These two sentences in the manuscript seem to oversimplify the inherent challenges of out-of-sample predictions by narrowly focusing on the perceived limitations of ANN models and potentially overestimating the capabilities of SVMs in mitigating bias, failing to acknowledge that the issue of generalization to unseen data is a pervasive challenge across all predictive models, not confined to ANN alone. A more nuanced comparison considering the strengths and weaknesses of both approaches, along with strategies to bolster generalization capabilities (like cross-validation, regularization, feature engineering, etc.), would provide a more balanced and insightful introduction.

3.     Page 11 Line 263: Here the author mentioned non-landslides samples. It’s recommended for authors to describe their non-landslide sampling strategies as this will affect the model results.

4.     Page 11 Lines 264-265: In the paper, the author's utilization of CNN and RNN models for tabular data seems to miss the mark, as the fundamental strengths of these models in recognizing spatial patterns (for CNN) and analyzing sequential data (for RNN) are not being leveraged. The data preprocessing to fit these models appears to be a superficial adaptation, transforming each sample to a format compatible with CNN and RNN, but not fully utilizing the potential of these models in a context where spatial or sequential information is critical. Therefore, it would be prudent to reconsider the choice of models or to justify the choice with additional information on how these models can enhance the analysis beyond traditional methods suitable for tabular data.

5.     It was observed that the input data mixes categorical and continuous variables. Tree-based models like XGBoost and RF can manage this data naturally, but CNN and RNN models often require preprocessing, like one-hot encoding or embedding, for categorical data. It is recommended for authors to describe if any preprocessing was done for the categorical data and if the continuous variables were normalized or scaled before analysis with CNN and RNN models.

6.     In addition, was parameter tuning performed for all the models used in the study? If not, could the authors explain how the performance comparison was conducted relatively?

Overall, I believe the utilization of SAR for analyzing landslide inventory and its subsequent comparison with the LSM model results is commendable. However, the inclusion of CNN and RNN models seems to have detracted from the overall quality of the paper, as their application appears not well-suited to the data at hand and potentially undermines the robustness of the study’s findings.

Round 2

Reviewer 3 Report

Other comments have been well addressed. However, in your revised manuscript, where is the changes compared with the first version? Have you modified? Please modified your manuscript based on the comments given in the first round. After carefully revision of the discussion, this manuscript could be considered its potential of publication.

Reviewer 4 Report

The author has addressed the review comments. However, for Table 8, please consider removing or changing the row for complexity comparison, as the complexity of neural networks depends on the number of trainable parameters and model architectures, etc. 
